# Y-Site Physical Compatibility of Numeta G13E with Drugs Frequently Used at Neonatal Intensive Care

**DOI:** 10.3390/pharmaceutics12070677

**Published:** 2020-07-18

**Authors:** Katerina Nezvalova-Henriksen, Niklas Nilsson, Camilla Tomine Østerberg, Vigdis Staven Berge, Ingunn Tho

**Affiliations:** 1Oslo Hospital Pharmacy, Rikshospitalet/Ullevål, Hospital Pharmacy Enterprise, South Eastern Norway, 0050 Oslo, Norway; niklas.nilsson@sykehusapotekene.no (N.N.); vigdis.staven.berge@sykehusapotekene.no (V.S.B.); 2Department of Pharmacy, University of Oslo, 0316 Oslo, Norway; camilla_2222@hotmail.com

**Keywords:** patient safety, parenteral nutrition PN, paracetamol, vancomycin, fentanyl, precipitation, emulsion stability, PFAT5, paediatrics, clinical pharmacy

## Abstract

Preterm neonates require parenteral nutrition (PN) in addition to intravenous drug therapy. Due to limited venous access, drugs are often co-administered with PN via the same lumen. If incompatible, precipitation and emulsion destabilization may occur with the consequent risk of embolism and hyper-immune reactions. Information on intravenous compatibility is scarce. Our aim was to analyse the compatibility of Numeta G13E with paracetamol, vancomycin and fentanyl because of the frequency of their use. A panel of methods was chosen to assess precipitation (sub-visual particle counting, turbidity measurement, Tyndall beam effect and pH measurement) and emulsion destabilization (mean droplet diameter measurement and sub-visual counting of oil droplets, followed by estimation of PFAT5 (percentage of fat residing in globules larger than 5 µm) and pH measurement). Samples in clinically relevant mixing ratios were tested immediately and after 4 h. All samples of drugs mixed with Numeta G13E were compared to unmixed controls. None of the tested drugs precipitated in contact with Numeta G13E, and we did not see any sign of emulsion destabilization when clinically relevant mixing ratios were applied. These results are reassuring. However, when contact time exceeds the established norm, caution in the form of filter utilisation and close inspection is advised.

## 1. Introduction

Uninterrupted nutrition is of paramount importance in all preterm neonates and infants in the neonatal intensive care unit (NICU). This is because inadequate nutrient supplies are associated with extra-uterine growth restriction, increased frequency and severity of postnatal medical complications resulting from impaired immunity, suppressed motor- and neurodevelopment and severe retinopathy of prematurity [1,2]. Due to temporary gut immaturity, most preterm neonates require parenteral nutrition (PN) as their main nutrient supply, particularly in the immediate postnatal period [3]. However, providing optimal nutrition to this patient group remains a challenge. NICU patients receive 20% less PN than assumed due to a very complex clinical environment [1]. Because of the complex nature of PN containing lipids, amino acids, carbohydrates and added vitamins, trace elements and electrolytes, physical stability is a delicate balance to maintain. It is desirable that PN is administered via a separate catheter lumen to achieve this. Most preterm neonates have a single-lumen central venous catheter (CVC), in the best of cases a three-lumen CVC, or peripherally inserted central catheter (PICC), which then precludes the simultaneous administration of vital drugs, blood transfusions, blood sampling or central venous pressure monitoring [3]. Pausing the nutrient supply and flushing the i.v. lines prior to and after drug administration is undesirable due to hypervolemia and low fluid capacity. Therefore, drugs are often administered simultaneously with PN via the same catheter lumen after all. This must be regarded as off-label use as co-administration rarely is described in the summary of product characteristics (SmPCs). Co-administration is known to increase the risk of incompatibility reactions between the products [4]. The consequences of incompatibilities between PN and drugs include crystalline particle formation and lipid emulsion destabilization, which may in turn lead to lumen occlusion, oxidative stress, organ defects and, in worst case scenario, emboli formation [4,5,6]. A tragic example of incompatibility with a lethal outcome are neonatal deaths following concurrent intravenous administration of ceftriaxone and calcium [7]. Both the EMA (European Medicines Agency) and FDA (U.S. Food and Drug Agency) express specific concerns regarding co-administration and incompatibility in their guidelines for the development of drugs to neonates [8,9].

Unfortunately, information about which drugs may be compatible with PN during Y-site administration is very scarce. Staven et al. performed several compatibility tests, assessing both visual and sub-visual particles/fat droplets and found that ampicillin, fosphenytoin and furosemide precipitated when mixed with Olimel N5E, Numeta G16E and a locally compounded preterm mix from Fresenuis Kabi, whereas ceftazidime, clindamycin, dexamethasone, fluconazole, metronidazole, ondansetron and paracetamol were compatible with Olimel N5E and Numeta G16E [10]. However, for the preterm mix, an unexpected micro-precipitation, probably caused by an interaction between copper and cysteine, disturbed the analyses so that none of the drugs could be concluded as compatible [11]. No emulsion destabilization was noted [10,11]. Fox et al. found that caffeine citrate, clindamycin, enalaprilat, epinephrine, fluconazole, fosphenytoin, hydrocortisone, metoclopramide and midazolam were compatible with a locally compounded neonatal lipid-free PN solution for up to 3 h in a simulated Y-site injection. Amiodarone, pentobarbital, phenobarbital, and rifampin were not compatible with the neonatal PN solution. Of note is the fact that only visual examination was performed [12]. Greenhill et al. reported no incompatibilities when mixing calcium gluconate, adrenaline, vasopressin and milrinone with another locally compounded lipid-free PN solution; however, this study was also carried out using drugs and dosages most commonly used in older paediatric patients [13]. Veltri and Lee found that neonatal lipid-free PN solutions with added amino acids were compatible with several drugs, including frequently used antibiotics, such as cefotaxime, penicillin G, and metronidazole, but were incompatible with acyclovir and ampicillin [14]. Both visual examination and Tyndall effect were utilised. Watson analysed the compatibility of 28 antibiotics with a locally compounded lipid-free PN solution for slightly older children weighing between 5 and 30 kg. He measured pH changes and used visual inspection. He found that ampicillin, cefamandole, cephalothin, cephradine and oxacillin led to a pH change in the PN solution (an increase) and ampicillin and cephradine produced a visible precipitate of calcium phosphate [15]. None of these studies were able to examine emulsion destabiliazation, since they excluded the lipid-phase. The results from these studies, whilst contributing to the information pool, are either not updated with respect to the parenteral nutrition currently used at NICUs or not generalisable when combinations with locally compounded PN solutions were tested. In addition, only Staven et al. performed a battery of compatibility tests that would ensure the reliability and reproducibility of their results [10]. 

The aim of our study was to use the same battery of compatibility tests to analyse the Y-site compatibility of Numeta G13E, to the best of our knowledge the only universal three-in-one PN mixture for premature neonates, with three drugs frequently administered together with it via the same catheter lumen: paracetamol, vancomycin and fentanyl. No documented compatibility information is available for such co-administration.

## 2. Materials and Methods

### 2.1. Materials

An overview of Numeta G13E, additives and drug formulations, dilution media and concentrations is presented in Table 1 and Table 2. European guidelines and information from products’ SmPCs were applied for maximum shelf-life after first opening or reconstitution of drugs, dilution media and PN after mixing and addition of supplements [16].

### 2.2. Selection of Test Materials

Numeta G13E is the three-in-one PN admixture used at our local NICU. An overview of the composition of Numeta G13E is shown in Appendix A.

Bedside observations of drugs used at the local NICU formed the base for selection of drugs to be investigated. Frequently used drugs with an infusion time of 15 min or more combined with pH dependent solubility were regarded as relevant. The final selection was done after discussions with the clinicians, taking their priorities into consideration. Paracetamol, vancomycin and fentanyl were selected for analysis of Y-site infusion with Numeta G13E with additives (from now on referred to as Numeta G13E+ in this manuscript). Trace elements, water-soluble and fat-soluble vitamins are always added to PN admixtures in NICU, and maximum permissible amounts for prematurely born infants of Peditrace (10 mL), Soluvit (3 vials) and Vitalipid infant (30 mL) (as stated by the manufacturer) were added in order to represent the extreme case scenario during co-administration via an intravenous catheter [17]. The composition of each type of additive is shown in Appendix A.

### 2.3. Study Design

To simulate the Y-site mixing ratios of Numeta G13E+ and the selected drugs, infusion rates of Numeta G13E and the selected drugs were utilised. The infusion rates for Numeta G13E were calculated based on the daily requirements using the ESPEN/ESPGHAN guidelines for paediatric parenteral nutrition [1], and the volume (mL/day), covering the nutritional requirements for several weight categories (kg) from 0.5 to 10 kg. It is unlikely that neonates in NICU should weigh 5.0 kg or more, but this weight class was included to represent a possible extreme. The 8 h and 24 h infusion regimens were used to calculate the infusion rates (mL/h); again, an 8 h infusion is too fast in the NICU-setting, but was necessary to represent an extreme situation. The drug doses were based on local guidelines and information in the British National Formulary (BNF) for children [18], the concentrations were chosen based on dialogue with clinicians to ensure clinical relevance and mL/kg was calculated for the same weight categories as for Numeta G13E. The infusion rates of the drugs were obtained by observation at NICU and when necessary calculated using the local guidelines at Oslo University Hospital that are based on recommendations in Neofax, BNF for children, Pediatric and Neonatal Dosage Handbook [18,19,20] and the Norwegian Medicines for Children Network’s mixing tables [21]. To simulate a range of potential mixing ratios of Numeta G13E and paracetamol, vancomycin and fentanyl, respectively, that might occur in the infusion line, mixing ratios were calculated by dividing the infusion rate of each drug with the infusion rate of Numeta G13E for each weight category (Figure 1, page 5). The most extreme ratio of Numeta G13E+ > drug as well as the 1 + 1 ratio were chosen. In cases where no ratio of drug > Numeta G13E+ was identified, two mixing ratios with more Numeta G13E+ relative to the drug in question were chosen instead, as shown in Table 3.

### 2.4. Sample Preparation

To enable potential precipitate detection, Milli-Q water was used to substitute the lipid constituent of Numeta G13E as suggested by Staven et al. [22]. This admixture will be referred to as *aq*Numeta G13E+ in this manuscript. Only trace elements were added to *aq*Numeta G13E and no vitamins because the water-soluble vitamins discolour the solution and may lead to analytical problems and lipid soluble vitamins are insoluble in water (Table 4).

To test emulsion stability, Numeta G13E+ was mixed with both trace elements and vitamins. Maximum amounts were added according to manufacturers’ recommendations to mirror the extreme case scenario in clinical practice (Table 4).

Samples for analysis were all prepared in a laminar airflow hood under ambient laboratory conditions by adding *aq*Numeta G13E+ or Numeta G13E+ to the drug in the three selected mixing ratios. Control samples (unmixed drug and PN) were also prepared in the laminar airflow hood. Samples for precipitate testing (where *aq*Numeta G13E+ was added to the drug) were mixed in 100 × 24 × 1.0 mm flat-bottomed glass tubes (Scherf Präzision Europa GmbH, Germany) for test of Tyndall effect and in sterile 50-mL polypropylene tubes (Corning, Mexico) for particle counting, turbidity and pH measurements. All solutions (not the emulsion) were filtered directly into the container through a sterile 0.22 μm syringe filter (VWR, Radnor, PA, USA) to reduce background particle pollution. In total, six parallels were prepared, each containing the calculated ratio of drug and *aq*Numeta G13E+: three for immediate testing and three for testing after 4 h. Since some of the analyses are destructive, the four-hour analyses could not be performed on the same sample as the immediate. Samples were kept at room temperature until they were analysed. Controls containing Milli-Q water, drug only and *aq*Numeta G13E+ only were also prepared. For turbidity measurements, the samples were transferred to the test glass and the outside was wiped with a glass wipe to remove dust that could influence the measurements.

Likewise, samples for emulsion stability testing were mixed in sterile 50-mL polypropylene tubes (Sigma-Aldrich Química, Toluca, Mexico). Three parallels containing each calculated ratio of drug and Numeta G13E+ (used for both immediate and four-hour analyses), as well as two controls containing Numeta G13E+ were used.

### 2.5. Analyses

A panel of quality-assured methods and established acceptance criteria was used to assess potential particle formation and emulsion destabilization [22]. Samples were tested immediately after mixing, which in practical terms means within 1 h, and 4 h after mixing. The late time point was added to check for incompatibility that might occur at long contact times due to low infusion rates. All samples were compared with controls described in the section above. With the exception of visual examination and emulsion testing, different samples were required because the tests were destructive.

#### 2.5.1. Methods and Assessment Criteria for the Detection of Potential Particle Precipitation

Sub-visual particle counting was carried out by light obscuration (Accusizer Syringe Injection Sampler, Optical Particle Sizer, PSS NICOMP, Billerica, MA, USA) to find the total number of particles/mL ≥ 0.5 μm, 5 μm, 10 μm, and 25 μm. The sensor was used in summation mode. The accepted background count of Milli-Q-water and sampling tubes was set to be below 100 particles/mL ≥ 0.5 µm. A 15 mL sample was measured undiluted to avoid dissolution of potential precipitate. Criteria: not more than a total of 2000 particles/mL ≥ 0.5 μm [22], and larger particles not exceeding the limits for “large volume parenterals” (not more than 3 particles/mL ≥ 25 μm and 25 particles/mL ≥ 10 μm) [24]. Furthermore, the total number of particles ≥ 5 μm was counted because neonatal capillaries are of approximately that diameter [25].

Turbidity measurement (2100Qis Turbidimeter, Hach Lange GmbH, Duesseldorf, Germany). Criteria: the upper limits are 0.2–0.3 Formazine Nephelometry Units (FNU) [22].

Visual examination against a black background using the fiber optic Tyndall beam (Schott KL 1600 LED, Germany) and red laser pen (630–650 nm, P 3010 RoHS, Chongqing, China). Criteria: no visible signs of precipitation or Tyndall effect [22,26].

pH measurement was carried out using a pH meter (Seven Compact, Mettler Toledo, Greifensee, Switzerland). Criteria: a change in pH >1.0 pH unit could induce the risk of precipitation of a drug and pH > approximately 7.2 could induce the risk of calcium phosphate precipitation [27].

#### 2.5.2. Methods and Assessment Criteria for the Evaluation of Emulsion Stability

The hydrodynamic diameter of the oil droplet and polydispersity index (PDI) of the droplet distribution were measured using dynamic light scattering (Zetasizer nano series, Malvern instruments, Malvern, UK). The Z-average mean size was used as a mean droplet diameter (MDD). Criteria: MDD < 500 nm [28]. Low PDI indicates narrow droplet size distribution; PDI below 0.2 may be regarded as representative of monodisperse samples.

Light obscuration was used to investigate potential droplet growth in the large diameter tail of the droplet sizes of the o/w emulsion. The instrument (Accusizer Syringe Injection Sampler, Optical Particle Sizer, PSS NICOMP, Billerica, MA, USA) was used in extinction mode and the lower detection threshold was 1.80 µm. Samples were diluted to ensure detection of single droplets one at the time. The counts of number of droplets with a diameter (D) > 2 µm, > 5 µm and > 10 µm was derived. The equivalent spherical volumes (ESV; cm^3^) of the oil droplets was calculated according to Equation (1) and total spherical volume (TSV; cm^3^) was calculated according to Equation (2). Finally, the weighted volume (WV) percentage of lipid droplets > 5 μm (PFAT5) [28], but also 2 µm (PFAT2) and 10 µm (PFAT10) were estimated from Equation (3).
(1)$$ESV=\frac{\pi \times D^{3}}{6}$$
(2)TSV=number of particles×ESV
(3)$$PFATX=\frac{\left[TSV\;\left(cm^{3}\right)\times Density\;\left(\frac{g}{mL}\right)\times Dilution\;factor\right]}{\left[sample\;volume\;\left(cm^{3}\right)\times Final\;oil\;composition\;\left(\frac{g}{mL}\right)\right]}$$

The density of oil used in the calculations was 0.92 g/mL, and the final oil composition was 0.030 g/mL (including oil from Vitalipid Infant). Criteria: PFAT5 < 0.40% [29].

pH measurement criteria: emulsion destabilization is more likely to occur at pH values < 5.5 [29].

## 3. Results

### 3.1. Detection of Potential Particle Precipitation

The physico-chemical characteristics of the controls used during particle precipitation detection are shown in Table 5. Sub-visual particle counts for *aq*Numeta G13E+ and paracetamol and vancomycin were low and not influenced by time. Tests of controls requiring large sample volume, such as particle counting (40 mL) and extended period of time (4 h), were not performed for fentanyl based on internal routines for drugs containing narcotic controlled substances. All control samples appeared clear upon visual inspection apart from a weak Tyndall effect in *aq*Numeta G13E+ samples, which is an inherent phenomenon that has been described earlier [26], and a weak Tyndall effect in vancomycin samples. All controls showed low turbidity. pH was in the range given in the SmPC for paracetamol and *aq*Numeta G13E+, but somewhat lower for fentanyl (Table 2 and Appendix A). For vancomycin powder for infusion no pH information after reconstitution was given in the SmPC, but after reconstitution in glucose 50 mg/mL (pH 3.5–5.5) the drug showed the most acidic pH at 3.2.

The results from precipitation testing are presented in Table 6. Deviations from acceptance criteria are shown in bold. Total particle counts above 0.5 µm, as well as counts for particles larger than 5 µm, 10 µm, and 25 µm, respectively (results not shown), were all low and within the acceptance criteria. The only drug that showed borderline signs of precipitation was fentanyl with FNU nearing 0.2 when *aq*Numeta G13E+ was present in abundance. However, these values did not differ greatly from the average FNU count found for the control samples containing only fentanyl. Other authors suggested using changes < 0.5 NTU (NTU is equivalent to FNU up to 40 NTU [30]) as acceptance limit [11]. All three drugs displayed occasionally weak Tyndall effects independent of mixing ratio with *aq*Numeta G13E+ and the time factor.

### 3.2. Evaluation of Emulsion Stability

The results from the emulsion stability analyses of mixed samples are presented in Table 7 with values exceeding the stipulated acceptance criteria or deviations shown in bold. The control sample of Numeta G13E+ showed mean droplet diameter below 270 nm with a narrow droplet distribution as indicated by low PDI. The PFAT5 of 0.2–0.3% confirmed that a low percentage of the lipid can be found in droplets with a diameter of 5 µm or larger.

For mixed samples, all mean droplet diameters (Z-average values) were slightly lower than the control sample and low PDIs suggested stable emulsions also after mixing with the respective drugs. For Numeta G13E+ with paracetamol, the PFAT5 values increased slightly both with increasing Numeta G13E+ volume in the mixing ratio and proportionately with contact time when Numeta G13E+ was in abundance: mixing ratio 1 + 10. When paracetamol and Numeta G13E+ were almost in equal proportions, in the ratio of 3 + 2, the values of PFAT5 were above the limit irrespective of time. Not surprisingly, the PFAT2 values (the percentage of lipid droplets > 2 μm: results not shown) displayed the same pattern and in the case of the 3 + 2 ratio, so did PFAT10 values (the percentage of lipid droplets > 10 μm: results not shown). pH of the mixed samples remained in the same range as the unmixed Numeta G13E+.

In the case of vancomycin, immediate contact with Numeta G13E+ in the mixing ratio 1 + 1 seemed to produce a value of PFAT5 above the stipulated acceptance criterium. This also occurred in the case of PFAT2 and PFAT10 values (results not shown). This was also noted when vancomycin was mixed with Numeta G13E+ in the ratio 1 + 2; however, here the time factor played a role because immediate contact did not yield PFAT5 values that deviated from the accepted normal. This was also the case for PFAT2 and PFAT10 values (results not shown). pH of the mixed samples was again in the same range as the unmixed Numeta G13E+, even though the vancomycin control sample showed a clearly more acidic pH (Table 5). Fentanyl was not found to have any effect on any of the PFAT values when mixed with Numeta G13E+. Again, no marked changes in pH values were noted for the mixed samples as compared to the unmixed Numeta G13E+ control.

## 4. Discussion

Our findings are reassuring for clinical use. In summary, none of the tested drugs precipitated when in contact with Numeta G13E, and we did not see any clear signs of emulsion destabilization when clinically relevant mixing ratios were applied.

No positive precipitation results were found when analysing sub-visual particle counts exceeding 0.5 μm/mL, turbidity and pH changes for paracetamol and vancomycin when these drugs were mixed with *aq*Numeta G13E+. These results are in accordance with those reported by Staven et al. [10] and Veltri and Lee [14]. A positive Tyndall effect was noted for paracetamol when mixed with *aq*Numeta G13E+ in equal amounts but only after 4 h contact time and as for the other mixing ratios, a Tyndall effect was seen immediately after mixing. Vancomycin also displayed a positive Tyndall effect when mixed with *aq*Numeta G13E+ both immediately and after 4 h in all but one mixing ratio where *aq*Numeta G13E+ was in abundance. However, the Tyndall effect was also observed in the unmixed controls of vancomycin (Table 5), and since the drug comes as a powder that requires reconstitution before use, the observed Tyndall effect might derive from dissolving the powder. The same has been observed for ampicillin in earlier studies [11]. All visual inspection methods are highly subjective so data interpretation can be challenging and in general not reproducible, hence, these results need to be interpreted with caution [26]. It can be noted that Staven et al. tested several generic paracetamol products with different excipient compositions and the generic products behaved differently in Tyndall light, where some products showed no Tyndall effect, whereas others had an inherent Tyndall effect and even displayed turbidity results > 0.4 FNU [10,22]. The composition of the paracetamol product in the current study was not similar to any of the products tested in the above-mentioned studies.

Fentanyl’s FNU counts came close to our acceptance limit of 0.2–0.3, but other researchers suggest a change between unmixed controls and the mixed samples < 0.5 as the acceptance limit. The fentanyl control (fentanyl not mixed with *aq*Numeta G13E+) displayed similar characteristics, and the effect of background noise must not be underestimated. The number of particles per mL exceeding 0.5 μm was low for all fentanyl mixed samples and well within the acceptance criteria, and no pH changes were noted when fentanyl was mixed with *aq*Numeta G13E+. No other studies performed turbidity measurements on fentanyl, so we lack a comparator. However, Veltri and Lee found fentanyl to be compatible with PN [14]. Furthermore the manufacturer provided test results on Numeta G13E mixed with fentanyl 3.6 µg/mL in the mixing ratio 1 + 10 (drug + PN) and found it compatible [17].

From a theoretical perspective, precipitation of paracetamol and fentanyl after mixing with the aqueous phase of the neonatal TPN mixture is unlikely. Both paracetamol and fentanyl have pKa-values on the basic side with 9.5 and 8.99, respectively [31,32], and the pH of 5.4 for *aq*Numeta G13E+ promotes solubility of both drugs. The pH of both drug products was measured to be close to that of *aq*Numeta G13E+, and since the amino acids provide buffer properties the pH of the mixed samples resembled that of the *aq*Numeta G13E+. Vancomycin is an amphoteric glycopeptide with several pKa-values (2.6, 7.2, 8.6, 9.6, 10.5 and 11.7), but it is freely soluble in water [31], and the particles detected are most likely due to slow dissolution of the powdered drug after reconstitution, as mentioned above.

There was no destabilization of the emulsion when paracetamol and Numeta G13E+ were mixed in equal amounts, neither immediately nor after 4 h. However, when Numeta G13E+ was in abundance, a common scenario in the clinical setting, the PFAT5 values exceeded the acceptance criterium after 4 h contact time. This is not of a great concern as infusions of paracetamol rarely exceed 4 h. This phenomenon was also observed when both paracetamol and Numeta G13E+ were mixed in a 3 + 2 ratio, seemingly irrespective of contact time; however, the PFAT5 value after immediate contact was borderline and only the value after 4 h contact time was obviously higher than the limit of 0.4%. It should be mentioned that the PFAT5 monograph of the USP is intended for injectable lipid emulsions and not for complex PN admixtures that contain electrolytes. The electrolytes carry charges, which may lead to temporary flocculation of the droplets (i.e., loosely connected droplet aggregates) and might explain somewhat large variations (SD) in PFAT5 for some of the drug+PN− samples. Since flocculates may be redispersed easily this is not as serious as the droplet growth that leads to coalescence and phase separation. The latter is not a reversible process.

In the case of vancomycin, immediate contact with Numeta G13E+, when in equal mixing ratios, resulted in an increased PFAT5 value. However, after 4 h, the PFAT5 value returned to normal. Whether this was a chance finding or the effect of time destabilizing the lipid emulsion is difficult to judge. Especially since the time factor seemed to play a role in the destabilization of the lipid emulsion when vancomycin and Numeta G13E+ were mixed in the 1 + 2 ratio. Just like in the case of paracetamol, vancomycin infusion time rarely exceeds 4 h so the deviating findings should not be of clinical significance. Fentanyl and Numeta G13E+ showed no signs of emulsion destabilization, irrespective of mixing ratio and time.

Since the solubility of the selected drugs is promoted by the pH of the neonatal TPN, the drugs may be regarded as low-risk for co-administration with Numeta G13E. Nevertheless, these drugs are frequently used in the NICU and it is important for the clinical environment to obtain documented information supporting their co-administration. Typical high-risk drugs for co-administration with the neonatal Numeta G13E would be drugs with low pKa-values that risk precipitating at the pH-value governed by the TPN. One well-known example is furosemide, which has pKa of 3.8 and has been shown to precipitate with the Numeta G16E [10], which is indicated for term born neonates and children up to 2 years of age.

On account of there being no generally accepted golden standard to how compatibility between i.v. fluids should be investigated, and the fact that neither the EMA [8] nor the FDA guidelines [9] offer any recommendations, our study was conducted using state of the art, validated methods that have been applied in similar studies in literature [10,12,13,22]. Our results should be interpreted with the following limitations and strengths in mind: Only one person was carrying out the analyses. This may lead to bias, particularly when highly subjective methods such as visual examination and the Tyndall beam effect were used. Due to the static nature of the test set-up when mixing in test tubes (glass and polypropylene), the dynamic Y-site interaction between Numeta G13E and the tested drugs in a clinical setting using syringe pumps could not be recreated. We have therefore no way of certifying that the interaction in a test tube mirrors the one between two flowing liquids in the lumen of an i.v. catheter. On the other hand, we utilized a battery of quality assured test methods and established acceptance criteria in order to make as objective conclusions as possible. Furthermore, background noise was measured and thereby controlled for. Lastly, but by far not of least importance, we utilized clinically relevant drug and PN combinations in clinically relevant mixing ratios as well as extreme ratios in an attempt to cover as many paediatric patient scenarios as possible.

## 5. Conclusions

Our findings are reassuring. Neither paracetamol, vancomycin nor fentanyl precipitated when in contact with Numeta G13E and the emulsion remained stable when clinically relevant mixing ratios were utilized. Deviations from particle number and stability acceptance criteria did occur albeit on an insignificant scale. However, when contact time between paracetamol or vancomycin or fentanyl and Numeta G13E exceeds the established norm, caution in the form of filter utilization and close inspection is advisable.

## Figures and Tables

**Figure 1 pharmaceutics-12-00677-f001:**
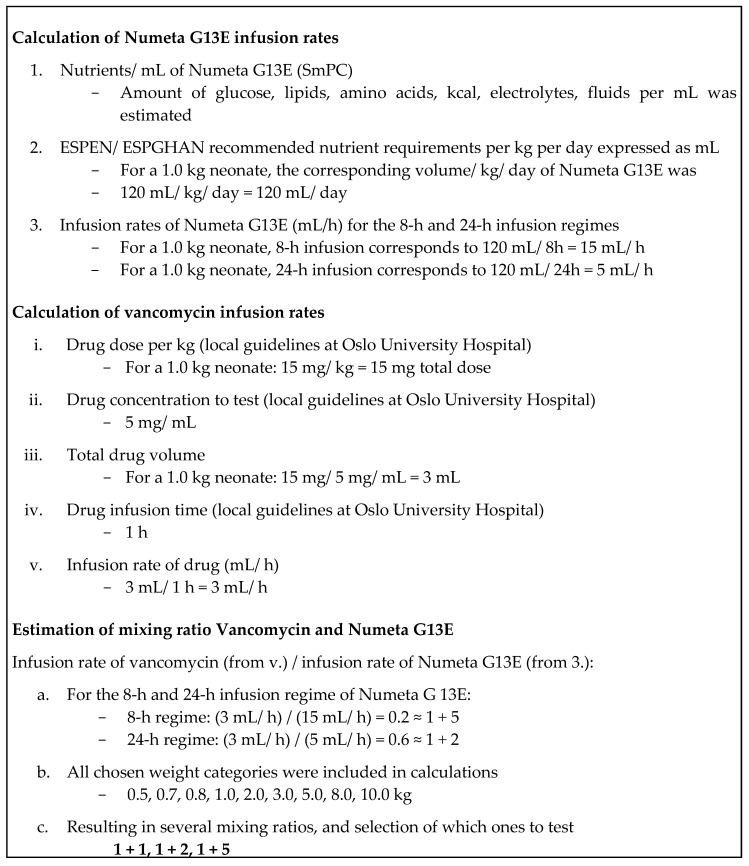
Example of the estimation of mixing ratios for Vancomycin and Numeta G13E.

**Table 1 pharmaceutics-12-00677-t001:** Overview of Numeta G13E and additives.

Product Type	Name	Manufacturer	Lot no.
Three-in-one PN admixture	Numeta G13E	Baxter	17E15N44 16K22N40
Trace elements	Peditrace	Fresenius Kabi	12LBL19 12LFL99
Vitamins—water soluble	Soluvit	Fresenius Kabi	10LF1840 10LK6141
Vitamins—lipid soluble	Vitalipid Infant	Fresenius Kabi	10LA5346 10LH3632

**Table 2 pharmaceutics-12-00677-t002:** Overview of drug formulations, dilution media, and their final concentrations.

Drug	Manufacturer	Lot no.	Dilution Medium	Concentration after Dilution
**Paracetamol** Excipients: mannitol, sodium citrate trihydrate, glacial acetic acid, aqua purificata pH: 4.5–5.5	B.Braun	17233450	Undiluted	10 mg/mL
**Vancomycin** Excipients: none pH: not stated	MIP	2725616	Glucose 50 mg/mL	5 mg/mL
**Fentanyl** Excipients: sodium chloride, hydrochloric acid/sodium hydroxide, aqua purificata pH: 5.0–7.5	Hameln	07400817A	Glucose 50 mg/mL	10 μg/mL

**Table 3 pharmaceutics-12-00677-t003:** Overview of selected mixing ratios of drug and Numeta G13E+ for Y-site simulation.

Drug	Selected Mixing Ratio Drug + Numeta G13E+
Paracetamol	1 + 1, 1 + 10, 3 + 2
Vancomycin	1 + 1, 1 + 2, 1 + 5
Fentanyl	1 + 1, 1 + 10, 1 + 20

**Table 4 pharmaceutics-12-00677-t004:** Composition of Numeta G13E after mixing and with all additives per L [23]. *aq*NumetaG13E+ refers to the aqueous phase used in tests for assessment of potential precipitation.

Component (Amount)	Numeta G13E+	*aq*Numeta G13E+
Olive oil (80%) (g)	17.4	-
Soy bean oil ^a^ (20%) (g)	13.0	-
Glucose monohydrate (g)	127.5	139.7
Alanine (g)	2.2	2.4
Arginine (g)	2.3	2.5
Aspartic acid (g)	1.6	1.8
Cysteine (g)	0.5	0.6
Glutamic acid (g)	2.7	3.0
Glycine (g)	1.1	1.2
Histidine (g)	1.0	1.1
Isoleucine (g)	1.8	2.0
Leucine (g)	2.7	3.0
Lysine (g)	3.0	3.3
Methionine (g)	0.6	0.7
Ornithine (g)	0.7	0.7
Phenylalanine (g)	1.1	1.2
Proline (g)	0.8	0.9
Serine(g)	1.1	1.2
Taurine (g)	0.2	0.2
Threonine (g)	1.0	1.1
Tryptophan (g)	0.6	0.6
Tyrosine (g)	0.2	0.2
Valine (g)	2.1	2.3
Sodium (mmol)	19.1	21.0
Potassium (mmol)	18.0	19.7
Magnesium (mmol)	1.4	1.5
Calcium ^b^ (mmol)	11.0	12.1
Phosphate ^c^ (mmol)	11.0	12.1
Acetate (mmol)	20.9	22.9
Chloride (mmol)	27.4	29.5
Malate (mmol)	9.3	10.2
Zinc (mg)	10.0	11.9
Copper (mg)	0.9	1.0
Manganese (mg)	0.04	0.05
Selenium (mg)	0.1	0.1
Fluoride (mg)	2.5	2.7
Iodide (mg)	0.04	0.05
Thiamine mononitrate (mg)	27.0	-
Riboflavin sodium phosphate (mg)	42.6	-
Nicotinamide (mg)	347.8	-
Pyridoxine hydrochloride (mg)	42.6	-
Sodium pantothenate (mg)	143.5	-
Sodium ascorbate (mg)	982.6	-
Biotin (mg)	0.5	-
Folic acid (mg)	3.5	-
Cyanocobalamine (mg)	0.04	-
α-Tocopherol (mg)	55.7	-
Retinol (mg)	6.0	-
Phytomenadione (mg)	1.7	-
Ergocholecalciferol (mg)	0.1	-

a: including soy bean oil from the addition of Vitalipid Infant, b: from calcium chloride dehydrate, c: from sodium glycerol phosphate and the lipid emulsion. Phosphate contribution from Vitalipid Infant is not known.

**Table 5 pharmaceutics-12-00677-t005:** Physico-chemical characteristics of unmixed controls of *aq*Numeta G13E+ and drugs (average values ± SD, *n* = 3).

Control	Particles/mL ≥ 0.5 μm		Turbidity (FNU)		Visible Particles or Tyndall Effect (+/−)		pH	
	0 h	4 h	0 h	4 h	0 h	4 h	0 h	4 h
*aq*Numeta G13E+	12 ± 5	25 ± 32	0.02	0.05	+	+	5.40	5.39
Paracetamol	9 ± 2	12 ± 9	0.03	0.01	−	−	5.24	5.23
Vancomycin	15 ± 9	8 ± 3	0.02	0.06	+	+	3.20	3.19
Fentanyl	N/A ^1^	N/A ^1^	0.16	N/A ^1^	−	−	4.81	N/A ^1^

^1^ N/A: Controls containing narcotic controlled substances are not routinely performed for large sample volumes or extended periods of time.

**Table 6 pharmaceutics-12-00677-t006:** Results from precipitation testing after mixing drug and *aq*Numeta G13E+ (bold font indicate values outside the acceptance criteria) (average ± SD; *n* = 3).

Drug	Mix Ratio	Particles/mL ≥ 0.5 μm		Turbidity (FNU)		Visible Particles or Tyndall Effect (+/−)		pH	
		0 h	4 h	0 h	4 h	0 h	4 h	0 h	4 h
Paracetamol	1 + 1	52 ± 26	78 ± 31	0.01 ± 0.01	0.01 ± 0.01	−	+	5.45 ± 0.02	5.47 ± 0.01
	1 + 10	18 ± 7	11 ± 2	0.08 ± 0.05	0.01 ± 0.02	+	−	5.45 ± 0.01	5.42 ± 0.01
	3 + 2	22 ± 4	9 ± 2	0.01 ± 0.01	0.05 ± 0.03	+	−	5.46 ± 0.04	5.45 ± 0.01
Vancomycin	1 + 1	36 ± 37	12 ± 5	0.00 ± 0.01	0.03 ± 0.03	+	+	5.41 ± 0.02	5.41 ± 0.01
	1 + 2	12 ± 2	12 ± 1	0.01 ± 0.02	0.02 ± 0.01	+	+	5.39 ± 0.01	5.39 ± 0.02
	1 + 5	18 ± 11	20 ± 10	0.01 ± 0.02	0.01 ± 0.01	−	−	5.45 ± 0.01	5.45 ± 0.01
Fentanyl	1 + 1	23 ± 3	16 ± 4	0.12 ± 0.00	0.14 ± 0.01	+	+	5.52 ± 0.01	5.50 ± 0.01
	1 + 10	20 ± 14	21 ± 5	**0.19 ± 0.02**	**0.17 ± 0.04**	−	−	5.39 ± 0.02	5.41 ± 0.01
	1 + 20	13 ± 3	18 ± 7	**0.17 ± 0.03**	0.14 ± 0.01	+	+	5.47 ± 0.02	5.44 ± 0.01

**Table 7 pharmaceutics-12-00677-t007:** Results from emulsion stability analysis when drug was mixed with Numeta G13E+ (bold font indicate values outside the acceptance criteria) (average ± SD; *n* = 3).

Drug	Mix Ratio	Z-Average (nm)	PDI	%PFAT5		pH	
				0 h	4 h	0 h	4 h
Numeta G13E+	-	**266 ± 1**	0.12 ± 0.01	0.20 ± 0.09	0.28 ± 0.08	5.50	5.50
Paracetamol	1 + 1	236 ± 1	0.15 ± 0.02	0.24 ± 0.08	0.34 ± 0.00	5.56 ± 0.04	5.54 ± 0.02
	1 + 10	240 ± 3	0.12 ± 0.01	0.24 ± 0.21	**0.66 ± 0.36**	5.49 ± 0.03	5.50 ± 0.03
	3 + 2	239 ± 1	0.12 ± 0.01	**0.41 ± 0.18**	**0.57 ± 0.41**	5.50 ± 0.05	5.47 ± 0.04
Vancomycin	1 + 1	241 ± 2	0.12 ± 0.04	**0.49 ± 0.22**	0.29 ± 0.10	5.50 ± 0.02	5.49 ± 0.02
	1 + 2	238 ± 4	0.13 ± 0.02	0.16 ± 0.02	**0.46 ± 0.35**	5.48 ± 0.01	5.47 ± 0.02
	1 + 5	240 ± 1	0.12 ± 0.01	0.17 ± 0.03	0.25 ± 0.05	5.52 ± 0.01	5.50 ± 0.01
Fentanyl	1 + 1	241 ± 2	0.10 ± 0.00	0.16 ± 0.04	0.09 ± 0.04	5.58 ± 0.03	5.59 ± 0.05
	1 + 10	240 ± 1	0.10 ± 0.01	0.13 ± 0.04	0.12 ± 0.04	5.58 ± 0.10	5.54 ± 0.03
	1 + 20	221 ± 1	0.12 ± 0.03	0.23 ± 0.06	0.16 ± 0.03	5.53 ± 0.02	5.47 ± 0.02

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
