# Peer review of "Y-Site Physical Compatibility of Numeta G13E with Drugs Frequently Used at Neonatal Intensive Care"

_pharmaceutics, 2020, doi:10.3390/pharmaceutics12070677_

Round 1

Reviewer 1 Report

I value the effort as described by the authors. 

i only have two reflections, 

first, how does the current method approach compare to the ema or fda guidelines on this topic ? 

second, the selection of 3 drugs is only poorly supported, but likely a 'logistics' related decision (paracetamol, vancomycin, fentanyl). How do these intermittent drugs reflect the 'physico-chemical' characteristics of intermittent drugs administered to neonates, like pKa ? and to what extent can high risk or low risk drugs (based on their physicochemical characteristics) be identified ? 

ceftriaxone is an obvious example of the relevant of this type of research in neonates, so that i suggest to add this to the paper ? 

Author Response

Dear Editor,

Thank you for considering our manuscript entitled “Y-site Physical Compatibility of Numeta G13E with Drugs Frequently Used at Neonatal Intensive Care” as an original research paper in the Pharmaceutics Special Issue on Paediatric Drug Delivery.

We thank the reviewers for their helpful comments and feedback of our manuscript, the responses to which are outlined below, including references to manuscript line numbers where required. All changes have been highlighted in yellow.

Kind regards,

Katerina Nezvalova-Henriksen and Ingunn Tho,

on behalf of the research team

Response to Reviewers' Comments:

Reviewer #1:

Comments and Suggestions for Authors:

I value the effort as described by the authors. 

The authors would like to thank the reviewer for the insightful comments. We have addressed the issues raised as outlined below.

i only have two reflections, first, how does the current method approach compare to the ema or fda guidelines on this topic ? 

Both EMA and FDA Guidelines for the development of drugs for the neonatal population mention specific concerns regarding co-administration and incompatibility, but none propose methods or approaches to investigate this complex matter (see statements below). The fact that both guidelines are vague, emphasizes the challenges experienced in clinical praxis and the lack of documented compatibility data. Our study is conducted in line with state-of-the-art research papers in the field and offers documented information for three frequently used drugs with a standard parenteral nutrition for neonates; information that is not available from the manufacturers.

A section has been added to the manuscript regarding the guidelines and approaches to investigate this topic (lines 55-57 and lines 381-382). The two guidelines have been added to the references.

EMA Guideline “Guideline on pharmaceutical development of medicines for paediatric use” EMA/CHMP/QWP/805880/2012 Rev. 2, dated 1 August 2013 states the following:

Neonates may only accept very small volumes of medicines in order to avoid volume overload and to allow sufficient room for essential fluid nutrition. Infusions must not be so concentrated that the appropriate dosing rates are not feasible by using standard pump equipment. These aspects should be considered during pharmaceutical development of all parenteral preparations intended for neonates, and in particular of those intended to be administered as a continuous infusion. In addition, specific concerns related to the incompatibility of the medicinal product with other co-administered medicinal products in the infusion line, osmolarity, inappropriate diluents, and potential for over- or under-dosing due to lag-volume effects in iv fluid lines should be investigated.”

FDA Draft Guidance “General Clinical Pharmacology Considerations for Neonatal Studies for Drugs and Biological Products Guidance for Industry” dated July 2019 states:

“Studies of drugs in neonates should account for potential interactions with tubing used for both parenteral and enteral administration and any potential interactions with co-administered fluids (including parenteral nutrition), enteral nutrition, and other therapeutic products.”  

second, the selection of 3 drugs is only poorly supported, but likely a 'logistics' related decision (paracetamol, vancomycin, fentanyl). How do these intermittent drugs reflect the 'physico-chemical' characteristics of intermittent drugs administered to neonates, like pKa ? and to what extent can high risk or low risk drugs (based on their physicochemical characteristics) be identified ? 

Thank you for this comment, which allows us to elaborate on the selection of the drugs and use of their physico-chemical characteristics for theoretical predictions.

The drugs were chosen because of their frequent use in NICU as stated in the manuscript. After bedside observations in NICU, frequently used drugs were discussed with the clinical pharmacist and nurses to make the priority of drugs for experimental investigations together with the most frequently used TPN for this population, Numeta G13E. The priority was given to drugs with infusion times of 15 minutes and above and drugs with pH-dependent or low aqueous solubility. The local NICU wanted to prioritize pain medication (paracetamol and fentanyl) and antimicrobial medication, where vancomycin was high in demand at the time of selection. Amendments have been made to the text to clarify this point (lines 104-107).

The table below summarizes some physico-chemical characteristics of the three selected drugs. Both paracetamol and fentanyl have high pKa values and pH dependent solubility. The products’ pH-values are on the weak acidic side promoting higher solubility. Vancomycin is an amphoteric glycopeptide with several pKa-values ranging from 2.6 to 11.7 (Table 1 below). The SmPC did not provide any information on the pH in the vancomycin product, but it was measured to be 3.2 in our studies. A new section has been added to the discussion to provide the theoretical perspective related to the physico-chemical characteristics of the drugs (lines 342-350)

Table 1: Physico-chemical characteristics of the drugs studied in the manuscript

Drug

pKa

Measured pH (control)*

Solubility

Reference

Paracetamol

9.5 (1)

5.2

Very sparingly soluble in cold water, but higher in hot water (1)

14 mg/ml at 25 ºC (2)

1)     Clarke’s Analysis of Drugs and Poisons via medicinescomplete.com

2)     PubChem

3)     Roy and Flynn, Pharm. Res. 1989; 6(2):147-51

Vancomycin

2.6, 7,2, 8.6, 9.6, 10.5,
11.7 (2)

3.2

1 in 10 of water (1)

0.225 mg/ml (2)

Fentanyl

8.99; 35 ºC (3)

4.8

0.2 mg/ml at 25 ºC (2)

0.74 mg/ml at 35 ºC and pH 7.0 (3)

* values from Table 5 of the manuscript

Since the solubility of the selected drugs is promoted by the pH of the neonatal TPN, the drugs may be regarded as low-risk for co-administration with Numeta G13E. Nevertheless, these drugs are in frequent use and the clinical environment was relieved to have documented information supporting their co-administration. Typical high-risk drugs for co-administration with the neonatal Numeta G13E (pH 5.4-5.5 in our studies) would be drugs with low pKa that risk precipitating at the pH-value governed by the TPN. One example is furosemide, which has pKa of 3.8, and has been shown to precipitate with Numeta G16E (Staven et al. J. Pharm. Pharmacol. 2017; 69: 448-462). It is well-known in our clinical environment that these drugs should not be co-administered with TPN. A new section has been added to further elaborate on the risk of the drugs to precipitate when mixed with Numeta G13E (lines 375-383).

ceftriaxone is an obvious example of the relevant of this type of research in neonates, so that i suggest to add this to the paper ? 

We, of course, agree with the reviewer that the tragic deaths of neonates after concurrent infusion of ceftriaxone and calcium is an obvious example of severe incompatibility reactions in neonates. This emphasizes the importance of this type of research and has been added to the manuscript together with the reference to the report by Bradley et al. Pediatrics 2009; 123: e609-e613 (lines 53-57).

Reviewer 2 Report

General comments:

The manuscript by Nezvalova-Henriksen et al. analyses the compatibility of the parenteral nutrition Numeta G13E with three drugs frequently used for premature neonates. Several methods were used to assess the physical stability of these preparations.

The manuscript is well written and organized.

Minors revisions/clarifications:

  • Line 114: Please, use the number “8” instead of “eight”, and use abbreviation “h” instead of “hours”;
  • Why did the authorsperform the analysis of the formulations after 4 hours? What are the infusion regimens? Were the mixed formulations (drug + Numeta) stored at room temperature after preparation?

Author Response

Dear Editor,

Thank you for considering our manuscript entitled “Y-site Physical Compatibility of Numeta G13E with Drugs Frequently Used at Neonatal Intensive Care” as an original research paper in the Pharmaceutics Special Issue on Paediatric Drug Delivery.

We thank the reviewers for their helpful comments and feedback of our manuscript, the responses to which are outlined below, including references to manuscript line numbers where required. All changes have been highlighted in yellow.

Kind regards,

Katerina Nezvalova-Henriksen and Ingunn Tho,

on behalf of the research team

Response to Reviewers' Comments:

Reviewer #2:

Comments and Suggestions for Authors

General comments:

The manuscript by Nezvalova-Henriksen et al. analyses the compatibility of the parenteral nutrition Numeta G13E with three drugs frequently used for premature neonates. Several methods were used to assess the physical stability of these preparations.

The manuscript is well written and organized.

We would like to thank the reviewer for the kind words. We have revised our manuscript as suggested by the reviewer.

Minors revisions/clarifications:

  • Line 114: Please, use the number “8” instead of “eight”, and use abbreviation “h” instead of “hours”;This has been revised throughout the manuscript.
  •  
  •  
  • Why did the authorsperform the analysis of the formulations after 4 hours? What are the infusion regimens? Were the mixed formulations (drug + Numeta) stored at room temperature after preparation?

Samples were analysed after 4 hours to check for incompatibility that occur at long contact times. This is relevant since infusion rates are slow for neonates. Moreover, repeating the analyses 4 hours after mixing is also in line with other reports in literature (e.g. Trissel). This has been clarified in the text (lines 203-205).

The mixed samples were stored at room temperature after preparation until they were analysed. This has been clarified in the manuscript (lines 156-157).
